# Melanosome Transport and Processing in Skin Pigmentation: Mechanisms and Targets for Pigmentation Modulation

**DOI:** 10.3390/ijms26178630

**Published:** 2025-09-04

**Authors:** Mengjing Bao, Mathias Gempeler, Remo Campiche

**Affiliations:** DSM-Firmenich, Wurmisweg 576, 4303 Kaiseraugst, Switzerland; mathias.gempeler@dsm-firmenich.com (M.G.); remo.campiche@dsm-firmenich.com (R.C.)

**Keywords:** melanosome transport, melanin degradation, melanogenesis, melanosome transfer

## Abstract

Achieving even skin tone and controlling pigmentation are key goals in dermocosmetics, given the impact of disorders like melasma, post-inflammatory hyperpigmentation, and age spots. The process of pigmentation begins with melanin synthesis within melanosomes, specialized organelles in melanocytes. Once produced, melanin is transferred to neighboring keratinocytes, where it forms protective caps over cell nuclei before undergoing eventual degradation. Disruptions at any stage of this complex process, whether in melanin production, melanosome transport, or post-transfer processing, can lead to visible pigmentation irregularities. While traditional treatments primarily focus on inhibiting melanin production (e.g., through tyrosinase inhibitors), emerging research highlights the important role of melanosome transport and keratinocyte-mediated processing in determining visible pigmentation. This review focuses on the underexplored stages of melanosome transport, transfer, and keratinocyte-mediated processing as promising targets for therapeutic and cosmetic strategies in managing pigmentation disorders.

## 1. Introduction

The pursuit of an even skin tone and effective pigmentation control is a major focus within the dermocosmetics industry, driving ongoing research into the complex mechanisms of melanogenesis and melanin regulation. This focus is especially important given the widespread impact of pigmentation disorders, such as melasma, post-inflammatory hyperpigmentation, and age spots (solar lentigines), which can significantly affect individuals’ quality of life [1,2]. Recent advances in understanding melanin dynamics have paved the way for innovative treatments targeting different stages of pigment formation and transfer.

Melanogenesis is a complex biochemical process that occurs within melanosomes in melanocytes [3,4]. As melanosomes mature, they are transported from the perinuclear region of melanocytes to the dendritic periphery and are tethered and anchored at dendritic tips [5]. Once positioned at the dendritic tips, melanin is transferred to keratinocytes by mechanisms that are still under investigation, potentially involving cytophagocytosis, vesicle shedding, exocytosis–phagocytosis or direct melanocyte–keratinocyte membrane fusion [5,6].

In keratinocytes, melanin forms supranuclear caps that protect DNA from UV-induced damage [7,8]. Within these cells, melanin is contained in “melanokerasomes”, specialized terminal compartments characterized by a single membrane, a weakly acidic lumen, and low hydrolase activity, which enable melanin to persist for weeks without significant degradation [9,10]. However, this stability is not permanent. Over time, melanin-containing compartments can be degraded via autophagy and lysosomal protease when specific pathways (Rab7B/42, cathepsins) are activated [11,12,13,14]. This degradation is more prevalent in light skin or under pharmacological induction [11,14,15,16,17].

While the role of active compounds in melanin synthesis is well established, and some studies have addressed aspects of melanosome transfer, the complete pathway—including intracellular transport, intercellular transfer, and post-transfer processing—remains less explored despite its critical importance for visible skin pigmentation [6,18,19]. Dysregulation of the transport and processing pathways can lead to uneven pigmentation, highlighting its relevance in both physiological and pathological conditions such as melasma, post-inflammatory hyperpigmentation, and pigmentary aging [20]. This review focuses on the often-overlooked aspects of transport and processing. Understanding these processes is vital for addressing pigmentation disorders and offers unique, underutilized targets for therapeutic interventions and cosmetic applications.

## 2. Pre-Transport Phase: A Brief Overview of Melanogenesis

### 2.1. Melanin Synthesis

Melanin synthesis is a tightly regulated, multi-step biochemical process involving a network of enzymes, transcription factors, and signaling pathways (Figure 1). The process is initiated by the oxidation of L-tyrosine to dopaquinone, catalyzed by tyrosinase (TYR), which is the rate-limiting enzyme in melanin production. Subsequent steps involve tyrosinase-related protein 1 (TYRP1) and dopachrome tautomerase (DCT/TYRP2), which further modulate the type and amount of melanin produced—eumelanin (brown-black) or pheomelanin (yellow-red) [3,4,18]. Recent studies highlight their cooperative roles in melanosome maturation. For example, TYRP1 stabilizes TYR and enhances its melanosomal localization, a process critical for efficient melanin synthesis [21].

At the molecular level, the microphthalmia-associated transcription factor (MITF), identified by Hodgkinson et al. in 1993, is the master regulator of melanogenesis [22]. MITF controls the transcription of TYR, TYRP1, and TYRP2, integrating signals from multiple upstream pathways, including the melanocortin-1 receptor (MC1R) pathway activated by α-melanocyte–stimulating hormone (α-MSH) [23,24]. Activation of MC1R by α-MSH increases intracellular cAMP, which activates protein kinase A (PKA). PKA phosphorylates and activates the cAMP response element-binding protein (CREB), which in turn enhances transcription of MITF [25]. MITF itself is phosphorylated at several sites, with Ser73 being a well-characterized phosphorylation site by MAPK/ERK signaling that promotes its transcriptional activity in melanogenesis [23,26]. Other critical pathways influencing MITF activity include WNT/β-catenin [27], and PKC signaling cascades [28].

Recent research has expanded the list of molecular regulators and biomarkers associated with melanogenesis. Proteins such as p53, hepatocyte nuclear factor 1α (HNF-1α), Sry-related HMg-Box gene 10 (SOX10), and paired box gene 3 (PAX3) modulate MITF expression, often in response to environmental cues like UV radiation [18,29,30]. Additionally, Nuclear factor erythroid 2-related factor 3 (NRF3), a recently identified transcription factor, coordinates melanogenesis by regulating cellular processes such as the macropinocytotic uptake of melanin precursors (L-Tyr and L-DOPA) and autophagosome-related genes for melanosome formation and degradation, linking metabolic state to pigment production [31].

### 2.2. Melanosome Maturation

Melanosomes undergo distinct stages of maturation during melanin synthesis (Figure 1). The early stage melanosomes (stages I and II) are unpigmented and characterized by the presence of structural protein such as premelanosome protein (PMEL, also known as GP100 or SILV), which forms the fibrillar matrix essential for subsequent melanin deposition [32,33]. Melanoma antigen recognized by T-cells 1 (MART1, also called Melan-A) is another critical biomarker for early-stage melanosomes. Its primary function is to form a complex with PMEL, thereby regulating PMEL’s expression, stability, trafficking, and proteolytic processing [34,35]. In addition to PMEL and MART1, other proteins such as ocular albinism type 1 (OA1 or GPR143) and the tetraspanin CD63 are involved in early melanosome biogenesis [36,37,38]. OA1 can interact with both MART1 and PMEL, and its distribution between melanosomes and lysosomes is regulated by ubiquitylation and ESCRT (endosomal sorting complex responsible for transport)-dependent sorting [36,39]. CD63 is required for the sorting of PMEL to intraluminal vesicles (ILVs) within early melanosomes, which is a prerequisite for the formation of the amyloid matrix that provides the scaffold necessary for melanin deposition [33,37].

*SLC24A5* encodes a potassium-dependent sodium/calcium exchanger (NCKX5) that is essential for melanogenesis through its role in maintaining ion homeostasis within melanosomes [40,41]. Knockdown of *SLC24A5* in human melanocytes significantly decreases pigmentation and leads to reduced protein levels of essential melanosome markers such as PMEL17, MART1, tyrosinase, and TYRP1, indicating *SLC24A5*’s role in early melanosome formation and maturation [41].

During the late stages of melanosome maturation—specifically stages III and IV—melanosomes undergo a series of biochemical and structural changes that lead to the synthesis and deposition of melanin. This maturation process is orchestrated by the delivery and activity of key melanogenic enzymes, including TYR, TYRP1, and DCT/TYRP2 [42]. These enzymes are translocated from early endosomal compartments to maturing melanosomes, where they catalyze the sequential reactions required for melanin biosynthesis.

The *OCA2* gene encodes the P protein, a melanosomal membrane protein that plays a crucial role in melanosome maturation. Its primary function is to regulate the ionic environment within melanosomes, particularly by contributing to a chloride ion current that helps maintain optimal melanosomal pH [43]. This pH regulation is essential for the optimal activity of TYR, thereby enabling effective melanin production and melanosome maturation [44]. Another player, solute carrier family 45 member 2 (*SLC45A2*), encodes a transmembrane transporter that plays a crucial role in the late stages of melanosome maturation [45]. It localizes predominantly to mature melanosomes and functions to increase melanosomal pH by exporting protons, and possibly glucose, thus creating a more neutral environment ideal for melanin synthesis [45,46]. SLC45A2 acts after OCA2 during melanosome maturation, maintaining the neutral pH initially established by OCA2 and supporting continued pigmentation in fully matured melanosomes [46].

Recent genome-wide screens have identified a host of additional genes implicated in melanosome maturation. Notably, the transcription factor Krüppel-like factor 6 (KLF6) has emerged as a critical regulator of late-stage melanosome maturation, with its loss resulting in a significant reduction in mature (stage IV) pigmented melanosomes [47]. KLF6 appears to regulate the expression of several melanogenic genes, including TYR, as well as genes involved in melanosome structure and function [47]. Additionally, COMMD3 (an endosomal trafficking protein) has been shown to regulate melanosomal pH during maturation. Loss of COMMD3 leads to abnormally acidic melanosomes, disrupting the transition from early (stage I/II) to mature (stage III/IV) stages. This impairment negatively affects melanin synthesis and overall melanosome development [47].

ATPase copper transporting α (ATP7A) is a copper transporter that plays a vital role in melanosome maturation by delivering copper directly to melanosomes, where it is essential for the enzymatic activity of TYR [48]. While TYR can acquire some copper in the trans-Golgi network, efficient and sustained activation of TYR within melanosomes requires direct copper supply by ATP7A [48]. The trafficking of ATP7A to melanosomes depends on the biogenesis of lysosome-related organelles complex-1 (BLOC-1), and defects in this pathway lead to mislocalization of ATP7A, impaired tyrosinase activity, and hypopigmentation, as observed in certain forms of Hermansky-Pudlak syndrome [48]. Thus, ATP7A is crucial for proper melanosome maturation and pigmentation by ensuring tyrosinase is adequately metalated within the melanosome.

## 3. The Transport Phase

Once melanosomes reach maturity, their transport journey begins. They are transported from the perinuclear region of melanocytes to the distal tips of dendritic extensions through a coordinated process involving both microtubule-based long-range transport and actin filament-mediated short-range movement [5] (Figure 2). Upon reaching the dendritic tips, mature melanosomes are transferred to adjacent keratinocytes, the primary cell type in the epidermis [5,6]. Within keratinocytes, melanosomes are internalized and processed in endolysosomal compartments, where melanin is ultimately stored in specialized structures [9].

### 3.1. Intracellular Transport of Melanosomes

#### 3.1.1. Microtubule-Based Long-Range Anterograde Transport

The microtubule-based long-range anterograde transport of melanosomes is fundamental to the proper distribution of pigment granules within melanocytes [49]. This process ensures that melanosomes are efficiently delivered from the perinuclear region to the cell periphery. Such distribution is vital for functions including photoprotection and the transfer of melanin to surrounding keratinocytes [5,50]. The transport mechanism is primarily mediated by microtubule-associated motor proteins, particularly members of the kinesin superfamily. Kinesin-1, composed of kinesin family member 5B (KIF5B) and kinesin light chain 2 (KLC2) subunits, has been identified as a principal driver of plus-end-directed, long-range melanosome movement along microtubules [51,52]. Key molecular biomarkers have been elucidated as essential components of this transport pathway. Rab1A, a small GTPase localized to mature melanosomes, plays a pivotal role in facilitating anterograde transport [49,51]. Rab1A recruits the adaptor protein SKIP (SifA and kinesin-interacting protein), which serves as a molecular bridge linking melanosomes to the kinesin-1 motor complex [51]. This interaction is critical for the selective and efficient attachment of melanosomes to kinesin-1, enabling their directed movement toward the cell periphery [51].

#### 3.1.2. The Switch of Melanosome from Microtubule to Actin Filament Networks

Melanosomes are subsequently handed over from the microtubule network to the actin cytoskeleton [53,54]. Following this handover, melanosomes undergo short-range anterograde transport along actin filaments toward the dendrite tips [55].

The transition of melanosomes involves a precisely coordinated switch from microtubule-based movement to actin filament-dependent positioning. Central to this process is the Rab27A–Melanophilin (Mlph)–Myo5A (Myosin–Va) tripartite complex, which acts as a molecular bridge between microtubule motors and the actin cytoskeleton [56,57]. Specifically, Rab27A, a Rab GTPase localized to melanosome membranes, recruits the adaptor protein Mlph, which in turn binds the actin motor Myo5A [57]. The phosphorylation state of Mlph can enforce track selection, biasing the Rab27A-Mlph-Myo5A complex toward either the actin or microtubule network, thus coordinating the switch between cytoskeletal systems [56].

The autophagy proteins LC3B (microtubule-associated protein 1A/1B-light chain 3 beta) and ATG4B (autophagy-related protein 4 homolog B) are directly involved in the switch of melanosomes from microtubule-based to actin-based transport [58]. LC3B facilitates the movement of melanosomes along microtubule tracks by enabling the assembly of the microtubule translocon complex on the melanosome membrane. At the microtubule-actin crossover junction, ATG4B enzymatically detaches LC3B from the melanosome membrane through delipidation, a step required for the transition of melanosomes onto actin filaments for short-range transport and subsequent transfer to keratinocytes [58].

The handoff between microtubule and actin networks could be facilitated by cytoskeletal crosslinkers. Spectraplakins, such as microtubule actin crosslinking factor 1 (MACF1), physically bridge microtubule and actin filament networks [59,60,61]. Similarly, scaffolding proteins like KANK1 have emerged as critical mediators of cytoskeletal crosstalk [62]. These findings suggest a potential role for such proteins in regulating melanosome transfer from microtubules to the actin cytoskeleton.

#### 3.1.3. Actin-Based Short-Range Transport

Actin-based short-range transport of melanosomes in melanocytes relies on a coordinated system of molecular motors, adaptor proteins, and cytoskeletal dynamics. Central to this process is still the tripartite complex comprising Rab27A, Mlph, and Myo5A, which facilitates melanosome movement along actin filaments [63,64,65,66].

Prohibitin (PHB) provides an additional mechanism for regulating this system. PHB stabilizes the interaction between Rab27A and Mlph independently of Myo5A, and its knockdown mimics the melanosome transport defects observed in Rab27A- or Mlph-deficient cells [67]. This underscores the complexity of actin-based transport, which integrates motor activity, adaptor protein scaffolding, and cytoskeletal remodeling.

#### 3.1.4. Microtubule-Based Long-Range Retrograde Transport

Retrograde transport of melanosomes in melanocytes refers to the movement of these pigment-containing organelles from the cell periphery back toward the perinuclear region along microtubules. This process is primarily mediated by the cytoplasmic dynein–dynactin motor complex, which moves cargo toward the minus ends of microtubules, typically oriented toward the cell center [68,69]. Melanosome attachment to this motor complex is facilitated by cargo receptors on the melanosome membrane, notably melanoregulin (Mreg), Rab36, Rab44 and Rab7 [68,70,71,72]. Specifically, Mreg interacts with the dynein–dynactin complex through Rab-interacting lysosomal protein (RILP) and the dynactin subunit p150 “Glued” (also known as DCTN1), facilitating the attachment of melanosomes to the motor machinery for centripetal movement [68]. Rab44 binds dynein-dynactin via its coiled-coil domain and localizes to mature melanosomes [70], and Rab36 functions partially redundantly with Mreg and Rab44 [71]. Notably, simultaneous depletion of Mreg, Rab36, and Rab44 nearly abolishes retrograde melanosome transport [70]. Rab7A also plays a key role by recruiting the dynein motor complex through its effector RILP, however, unlike Rab36 and Rab44, it primarily facilitates the transport of early and intermediate-stage melanosomes toward the cell center [72]. In mouse melanocytes that are deficient in Rab27A, which shows perinuclear aggregation of melanosomes, knockdown of Rab36 causes a stronger disruption of this aggregation compared to knockdown of Rab7 [71]. Together, Rab7A, Rab36, and Rab44 coordinate stage-specific pathways to ensure proper positioning of melanosomes within melanocytes.

The retrograde transport of melanosomes plays a critical role in maintaining melanocyte function and skin pigmentation by balancing organelle distribution [69]. This process may also indirectly enhance melanosome transfer to keratinocytes by retrieving melanosomes from the cell periphery, thereby enabling repeated cycles of dendritic extension and delivery. For example, Rab27A-deficient melanocytes show perinuclear melanosome aggregation, which is reversed by disrupting retrograde components like Mreg [68]. Additionally, knockdown of Dynlt3-a light chain subunit of cytoplasmic dynein-leads to peripheral accumulation of melanosomes, increased acidity of mature melanosomes, and reduced efficiency of their transfer to keratinocytes [69].

### 3.2. Intercellular Transfer of Melanocores or Melanosomes

Melanin transfer from melanocytes to keratinocytes has been proposed by four main models [5,6,50,73]. These models differ in whether they require direct physical contact between melanocytes and keratinocytes (Figure 3).

Two of the models—cytophagocytosis and membrane fusion—require direct cell–cell interaction. The cytophagocytosis model was visualized using electronic microscopy in the several studies [74,75]. In this model, the dendritic tips of melanocytes, which contain melanosomes, come into contact with the plasma membrane of keratinocytes. The keratinocytes then engulf these dendrite tips, forming a vesicle that contains the melanosomes [74,75]. Therefore, this model proposes that melanin is enclosed by three membranes: the intrinsic melanosome membrane, along with additional membranes derived from the plasma membranes of the melanocyte and the keratinocyte [74,76]. This vesicle fuses with lysosomes inside the keratinocyte, leading to the degradation of the surrounding membranes and the dispersion of melanin granules within the cytoplasm of the keratinocyte [77]. The membrane fusion model, on the other hand, involves the direct fusion of the plasma membranes of melanocytes and keratinocytes. Experimental support for this model has been provided by a primary research study, which used time-lapse digital imaging and electron microscopy of melanocyte–keratinocyte co-cultures [78]. The study demonstrated that filopodia or nanotubes (extensions of the melanocyte membrane) establish extensive contacts with keratinocytes, forming intercellular bridges through which melanosomes are transferred [78]. However, it cannot be excluded that these “bridges” are severed, and melanosomes are subsequently taken up via phagocytosis by keratinocytes, as has been observed in this study [79]. Additional evidence is needed to fully support this model.

In these cell-contact models, caveolae are crucial for melanin transfer between melanocytes and keratinocytes by regulating cell–cell interactions [80]. In co-culture models, caveolae concentrate at the melanocyte–keratinocyte interface, supporting the formation of dendrite-like protrusions needed for direct melanin transfer. Disruption of caveolae formation, such as through caveolin-1 (Cav1) depletion, impairs dendrite outgrowth and reduces cell–cell contacts in melanocytes, leading to significantly decreased melanin transfer to keratinocytes, despite an increase in melanin synthesis driven by upregulated cAMP signaling [80]. Filopodia-associated proteins, including β-catenin, Cdc42, Myosin X, and E-cadherin, are upregulated by ultraviolet radiation and Ca^2+^ stimulation and are important for filopodia formation and melanin transfer [81]. Rab17, which acts on melanosomes downstream of Rab27A, has also been reported to be required for melanocyte filopodia formation and thereby facilitates pigment transfer [82].

The other two models-exocytosis/phagocytosis and shedding vesicles-do not require direct physical contact between the two cell types. In the exocytosis and phagocytosis model, melanosomes become melanocores through a process in which mature melanosomes fuse with the melanocyte plasma membrane. This fusion leads to the exocytosis of the naked melanin core, called the melanocore, into the extracellular space [83]. This process is dependent on the small GTPase Rab11B and components of the exocyst complex, including Sec8 (gene name: EXOC4) and Exo70 (gene name: EXOC7) [84,85]. More recently, Rab3A has also been identified as a regulator of melanin exocytosis, particularly under stimulation by soluble factors from differentiated keratinocytes [86]. Keratinocytes then internalize these melanocores through phagocytosis, a process heavily dependent on actin cytoskeleton remodeling and regulated by Rho family GTPases such as Rac1 and Cdc42 [87]. Similarly, in the shedding vesicles model, melanocytes release vesicles loaded with melanosomes into the extracellular environment [88,89]. In a recent study using EGFP labeling and live imaging in a chicken embryonic skin model, researchers observed that melanosome transfer occurs through plasma membrane vesicles generated by melanocyte membrane blebbing. During this process, blebs encapsulate melanosomes, detach as vesicles, and are subsequently engulfed by neighboring keratinocytes [90]. Importantly, membrane blebbing and vesicle release were shown to depend on the activity of the Rho small GTPase [90].

## 4. Melanin Post-Transfer Processing in Keratinocytes

After melanin is transferred from melanocytes to keratinocytes, it undergoes a series of processing steps within the recipient cell. Once internalized—primarily through phagocytosis or micropinocytosis—melanin becomes enclosed in phagosomes that mature by fusing with lysosomes, forming specialized compartments known as melanokerasomes. These compartments are weakly degradative, allowing melanin to be retained rather than broken down. The melanokerasomes are then transported and positioned above the keratinocyte nucleus, where they form supranuclear caps that shield the nuclear DNA from UV radiation and contribute to the skin’s photoprotective barrier [5,9]. Transmission electron microscopy has revealed tethering structures linking melanokerasomes to the nuclear envelope, likely stabilizing their juxtanuclear position [9]. This post-transfer processing is essential for both the persistence of skin pigmentation and the protection of epidermal cells from UV-induced damage.

Melanin in keratinocytes is relatively long-lived and its levels diminish primarily due to the natural process of superficial keratinocyte sloughing as the epidermis renews itself approximately 40–56 days [10,91]. Most of the melanin transferred from melanocytes to keratinocytes is eventually lost from the skin surface as pigmented corneocytes are shed during this renewal cycle. Although some intracellular melanin degradation can occur, particularly more efficiently in lighter skin types, a significant portion remains intact until it is removed via epidermal turnover [11,15,92]. Nonetheless, understanding the mechanisms underlying melanin degradation within keratinocytes may offer valuable insights for developing effective skin-lightening strategies (Figure 4).

### 4.1. Melanin Uptake

Melanocore and melanosome uptake by keratinocytes is a highly regulated process involving distinct cellular pathways and specific receptors. The current literature shows that melanocores—the pigment-rich cores released from melanocytes—are primarily internalized by keratinocytes through phagocytosis, a process that depends on actin dynamics and is mainly regulated by small GTPases such as Rac1 and Cdc42 [87]. In contrast, whole melanosomes are mainly internalized by macropinocytosis, an actin-dependent process that predominantly involves CtBP1/BARS (C-terminal binding protein 1/brefeldin A ADP-ribosylated substrate) and small GTPase RhoA, indicating separate regulatory pathways for the two forms of melanin transfer [87].

A key receptor involved in melanin uptake is protease-activated receptor-2 (PAR-2) [93]. Studies have shown that the uptake of melanosomes by keratinocytes is dependent on PAR-2, either when keratinocytes are incubated with isolated melanosomes [94] or with pigment globules containing multiple melanosomes [88]. More recent studies have demonstrated that melanocore internalization triggers a stronger PAR-2 internalization response in keratinocytes than whole melanosomes, suggesting a receptor-mediated preference for melanocore uptake [87]. Additionally, keratinocyte growth factor (KGF/FGF7) and its receptor FGFR2b have been implicated in promoting melanosome uptake, particularly by enhancing the phagocytic capacity of keratinocytes, especially in lighter skin types [95,96,97]. Toll-like receptor 3 (TLR3) has also been reported to play a role in melanin uptake by keratinocytes. Upon stimulation—such as by the viral mimic poly(I:C), TLR3 activation in keratinocytes markedly enhances the internalization of melanosomes through endocytic mechanisms. This effect is mediated by the upregulation and activation of the small GTPases RhoA and Cdc42, which drive actin cytoskeleton remodeling necessary for efficient endocytosis. This highlights TLR3’s function as a molecular link between innate immune signaling and pigment transfer in the epidermis [98].

### 4.2. Melanin Retention and Degradation

Once within keratinocytes, melanin is typically surrounded by a membrane derived from the keratinocyte plasma membrane, especially in the case of melanocores, or by multiple membranes if internalized as part of a melanosome globule. Following internalization, melanin-containing compartments undergo intracellular trafficking, moving toward the juxtanuclear region of the keratinocyte. Here, the granules aggregate to form perinuclear caps—structures that function as protective "parasols" shielding the nuclear DNA from UV radiation [5,9].

Studies using mouse and human cells or skin tissues have demonstrated that the maturation of these melanin-containing vesicles is marked by a transition from early endosomal markers such as EEA-1 and Rab5 to late endosomal, such as Rab7, and lysosomal markers including CD63 and LAMP1, indicating their progression along the endolysosomal pathway [9,10,99]. Despite this fusion, melanin remains within a terminal lysosomal compartment—melanokerasome—that resists degradation, likely due to its weak acidity and low hydrolase activity [9].

Ultimately, the fate of melanin within keratinocytes involves a balance between storage for photoprotection and gradual degradation. Melanin granules are degraded through the autophagic pathway [11,14], and studies in human keratinocytes and skin models have shown that, once internalized, melanin-containing organelles in keratinocytes are targeted for selective degradation by the autophagy adaptor protein p62 [11,100]. The p62-labeled structures are then engulfed by autophagosomes marked by the presence of autophagy proteins like ATG7 and LC3 [11,100]. Rab7 has been reported to be involved in protein degradation within melanin-containing compartments, while having only a minor effect on melanin itself in keratinocytes [13]. This is likely because Rab7B/42 is required for the fusion of autophagosomes with lysosomes, a process that forms autolysosomes where partial degradation of melanin occurs [11,101]. A recent study conducted in human keratinocytes and skin models has identified cathepsin V (also known as cathepsin L2. Gene name: CTSV) as a key lysosomal protease involved in melanosome and melanocore degradation in keratinocytes [12]. Immunohistochemical studies have shown that cathepsin V is highly expressed across the epidermis in normal skin. Notably, cathepsin V expression levels were lower in the basal layer compared to the stratum corneum side in hyperpigmented regions. Furthermore, melanosome degradation was suppressed in cathepsin V knockdown cells, indicating its role in melanosome degradation [12].

## 5. Relevance to Cosmetic Dermatology and Therapeutics: Future Perspectives

From both therapeutic and cosmetic perspectives, a comprehensive understanding of the regulatory mechanisms underlying melanogenesis, melanosome maturation/transport, and degradation pathways is critical for developing targeted interventions in pigmentary disorders. To facilitate this, we have compiled a systematic list of key biomarkers associated with skin pigmentation (Table 1).

While most current skin-lightening agents focus on inhibiting melanin synthesis, relatively few target the full pathway of melanosome transport, and even fewer influence the degradation of melanin within keratinocytes [92,102,103]. Among the widely used skin-whitening actives, kojic acid, arbutin (beta-arbutin), azelaic acid, and vitamin C (ascorbic acid) primarily function by targeting tyrosinase, the rate-limiting enzyme in melanogenesis [102,104]. Kojic acid, a fungal metabolite, inhibits tyrosinase by chelating copper ions at its active site, thereby blocking the conversion of tyrosine to melanin and reducing pigmentation [105]. Arbutin, also known as β-arbutin, is a plant-derived β-D-glucopyranoside of hydroquinone that reversibly inhibits tyrosinase activity, leading to decreased melanin synthesis [106]. In contrast, α-arbutin, typically produced synthetically through enzymatic synthesis or microbial fermentation, is more stable and more effective than β-arbutin in skin-whitening applications [107]. Azelaic acid, a naturally occurring dicarboxylic acid, suppresses melanogenesis by inhibiting both tyrosinase and mitochondrial oxidoreductases, resulting in a lightening effect on hyperpigmented skin [108,109]. Vitamin C acts as an antioxidant and skin-brightening agent by directly inhibiting tyrosinase and reducing oxidized dopaquinone back to DOPA, thereby interfering with melanin production [110]. In contrast, niacinamide (vitamin B3) does not inhibit tyrosinase directly but reduces hyperpigmentation by blocking the transfer of melanosomes from melanocytes to keratinocytes, promoting a more uniform skin tone [111]. Collectively, these agents are commonly formulated in topical products to address hyperpigmentation, melasma, and uneven skin tone, predominantly by disrupting melanogenesis [108,111,112,113,114].

Targeting melanosome transport offers a novel and promising approach to modulating skin pigmentation. Unlike traditional skin-lightening agents that primarily inhibit melanin synthesis, such as tyrosinase inhibitors, this strategy directly influences the distribution of pigment across the skin without necessarily altering total melanin levels. Emerging evidence supports the potential of targeting melanosome transport (e.g., MNQO) as a viable strategy for pigmentation control. This process relies on specific proteins and signaling pathway, such as Rab27a, Myo5A, and melanophilin (Mlph), that regulate melanosome movement and can be selectively modulated to influence pigment transfer without altering melanin production [115]. Compounds such as centaureidin and pyruvate have been shown to alter cytoskeletal dynamics and inhibit dendrite formation in melanocytes, thereby reducing melanosome transfer and providing proof-of-concept for this mechanism [116,117]. Additionally, serine protease inhibitors that block PAR-2 activation have been found to induce depigmentation by disrupting melanosome distribution rather than melanin production [118]. Notably, wogonin, a flavonoid derived from *Scutellaria baicalensis Georgi*, and Metformin, have been shown to regulate both melanin synthesis and melanosome transport, further highlighting the potential for dual-action interventions in treating pigmentation disorders [19,119].

By regulating how melanosomes are transferred or moved, this opens up new therapeutic strategies that may be effective where traditional approaches fail, or that can be combined with existing treatments for enhanced results [118,120]. Modulating melanosome transport may also allow for more precise correction of pigmentation issues, possibly reducing the risk of hypopigmentation or irritation associated with more aggressive treatments that destroy pigment or skin cells [118,121]. Furthermore, some interventions have been shown to reversibly inhibit melanosome transfer without causing cytotoxicity, enabling adjustable and controlled pigmentation correction rather than permanent alteration [122].

In conclusion, exploring melanosome transport as a therapeutic target offers a promising new avenue for treating uneven skin pigmentation. By focusing on how pigment is distributed, rather than just how much is made, this approach could provide more precise, effective, and potentially safer solutions for a wide range of pigmentary disorders.

## Figures and Tables

**Figure 1 ijms-26-08630-f001:**
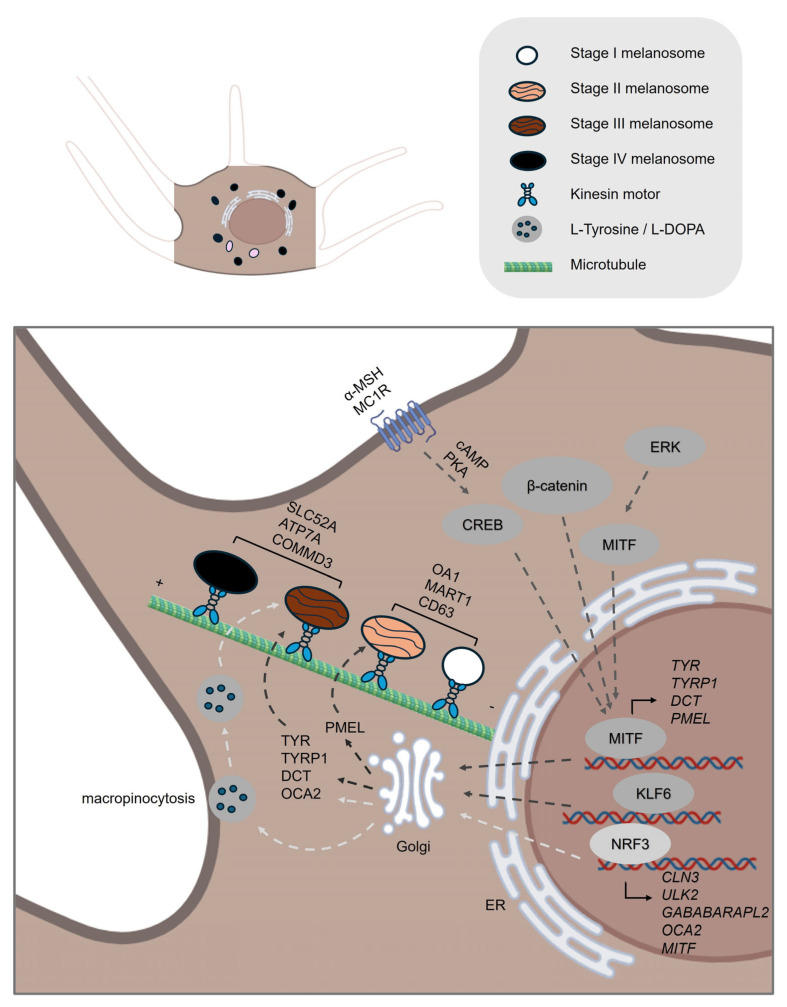
**Melanin Synthesis and Melanogenesis.** The key stages and molecular components of melanin synthesis and melanosome maturation in melanocytes. **Melanin Synthesis**: Initiated by tyrosinase (TYR), which oxidizes L-tyrosine to dopaquinone—a rate-limiting step. TYRP1 and DCT/TYRP2 further direct melanin type (eumelanin or pheomelanin) and stability. MITF, the master transcription factor regulating melanogenesis, is transcriptionally upregulated by the α-MSH–MC1R–cAMP–PKA–CREB and WNT/β-catenin signaling pathways. Additionally, MITF activity is modulated via phosphorylation by kinases such as MAPK/ERK. NRF3 also regulates precursor uptake (L-Tyr, L-DOPA) and autophagic pathways for melanosome dynamics. **Melanosome Maturation**: Stage I–II (Early Melanosomes): Unpigmented, scaffolded by structural proteins PMEL and MART1, which form the amyloid matrix. OA1 and CD63 contribute to vesicle sorting and PMEL trafficking. Stage III–IV (Late Melanosomes): Pigmented, with active melanin deposition catalyzed by TYR, TYRP1, and DCT. OCA2 regulates melanosomal pH via chloride ion transport; optimal pH is critical for TYR function. SLC45A2 maintains pH neutrality by exporting protons/glucose, acting downstream of OCA2. ATP7A supplies copper to melanosomes for TYR activation; trafficking is dependent on BLOC-1 complex. Transcription factor KLF6 and trafficking protein COMMD3 regulate late-stage maturation by controlling gene expression and pH balance, respectively. The schemes were created in BioRender. Bolis, M. (2025) https://BioRender.com/h42n006.

**Figure 2 ijms-26-08630-f002:**
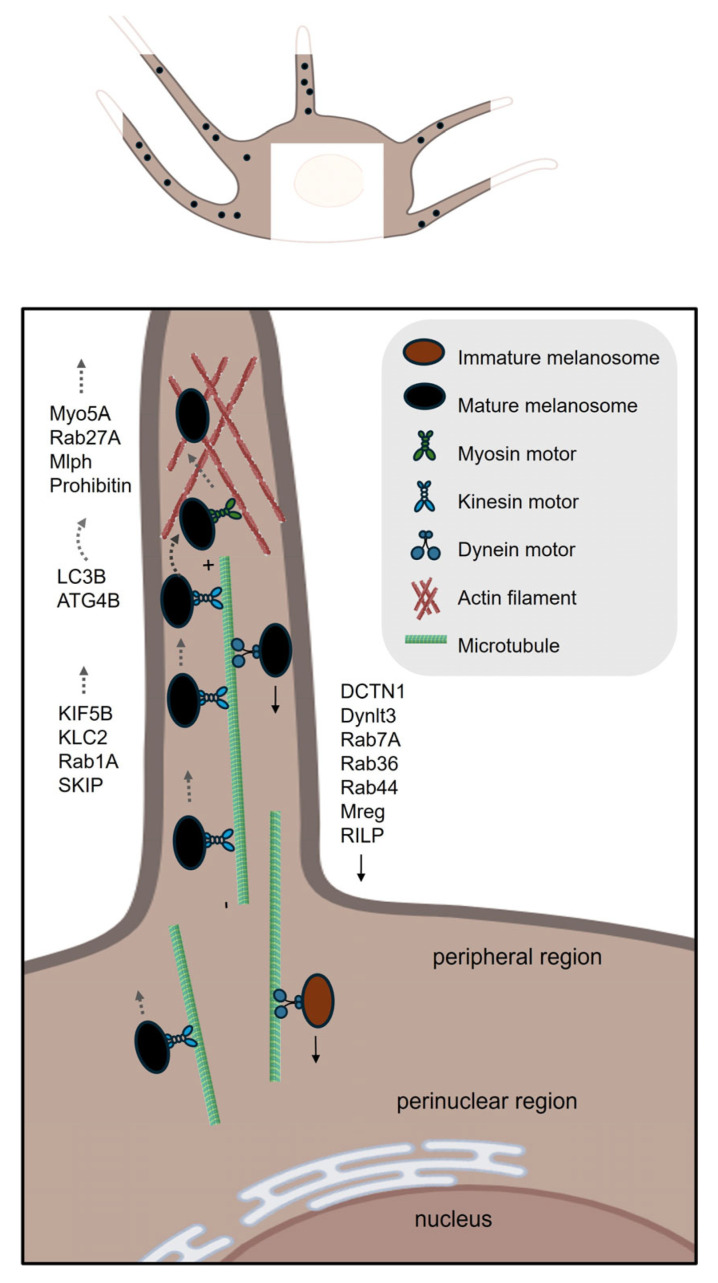
**Melanosome Intracellular Transport in Melanocytes.** This schematic illustrates the process that occurs from the perinuclear region towards peripheral and dendritic tips of melanocytes. The transport phase, detailing the intracellular transport mechanisms and molecular regulators involved in this multistep journey. **1. Long-Range Anterograde Transport (Microtubule-Based)**: Mature melanosomes move from the perinuclear area toward the dendritic tips of melanocytes along microtubules. Driven by kinesin-1 (KIF5B/KLC2) motors. Rab1A on melanosomes recruits SKIP (PLEKHM2), linking melanosomes to kinesin for transport. **2. Transition from Microtubule to Actin Filament Networks**: At the microtubule-actin interface, melanosomes are handed over to the actin cytoskeleton for final delivery to dendritic tips. Mediated by the Rab27A–Melanophilin (Mlph)–Myo5A complex. LC3B facilitates initial microtubule transport; ATG4B detaches LC3B to enable actin-based movement. **3. Short-Range Anterograde Transport (Actin-Based)**: Melanosomes are propelled to the dendrite tips via actin filaments, guided by the Rab27A–Mlph–Myo5A complex. Prohibitin (PHB) stabilizes the Rab27A–Mlph interaction, enhancing transport efficiency. **4. Long-Range Retrograde Transport (Microtubule-Based)**: Melanosomes are transported back to the perinuclear region via dynein–dynactin motor complexes. Mediated by Mreg, Rab7A, Rab36, and Rab44, which recruit the RILP–DCTN1 (p150^Glued) complex. Retrograde transport balances melanosome distribution and enables recycling for efficient pigment delivery. The schemes were created in BioRender. Bolis, M. (2025) https://BioRender.com/h42n006.

**Figure 3 ijms-26-08630-f003:**
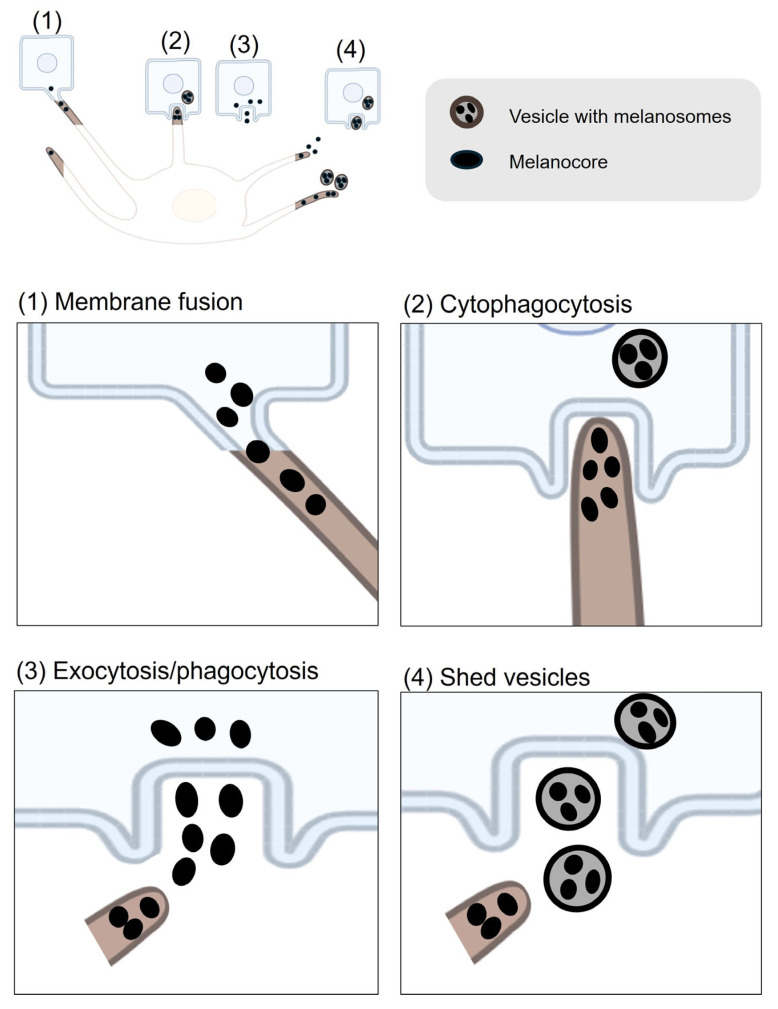
**Melanin Intercellular Transfer Mechanisms.** The process occurs between melanocytes and keratinocytes and requires direct or indirect interaction between the two cell types. Melanin transfer from melanocytes to keratinocytes occurs via four proposed models, distinguished by whether direct cell–cell contact is required. **1. Membrane Fusion (direct contact)**: Plasma membranes of melanocytes and keratinocytes fuse, forming conduits (e.g., filopodia or nanotubes) for direct melanosome transfer. Caveolae, particularly caveolin-1 (Cav1), regulate this interaction by supporting dendrite formation and cell–cell contact. **2. Cytophagocytosis (direct contact)**: Keratinocytes engulf melanocyte dendrite tips containing melanosomes, internalizing melanin into vesicles that fuse with lysosomes for melanin dispersion. **3. Exocytosis/Phagocytosis (no direct contact)**: Melanocytes exocytose melanocores into the extracellular space, which keratinocytes then phagocytose. This process is Rab11B-dependent in melanocytes and actin-regulated in keratinocytes via Rho GTPases. **4. Shedding Vesicles (no direct contact)**: Melanocytes release melanosome-containing vesicles into the extracellular space. Keratinocytes internalize these vesicles through macropinocytosis. The schemes were created in BioRender. Bolis, M. (2025) https://BioRender.com/h42n006.

**Figure 4 ijms-26-08630-f004:**
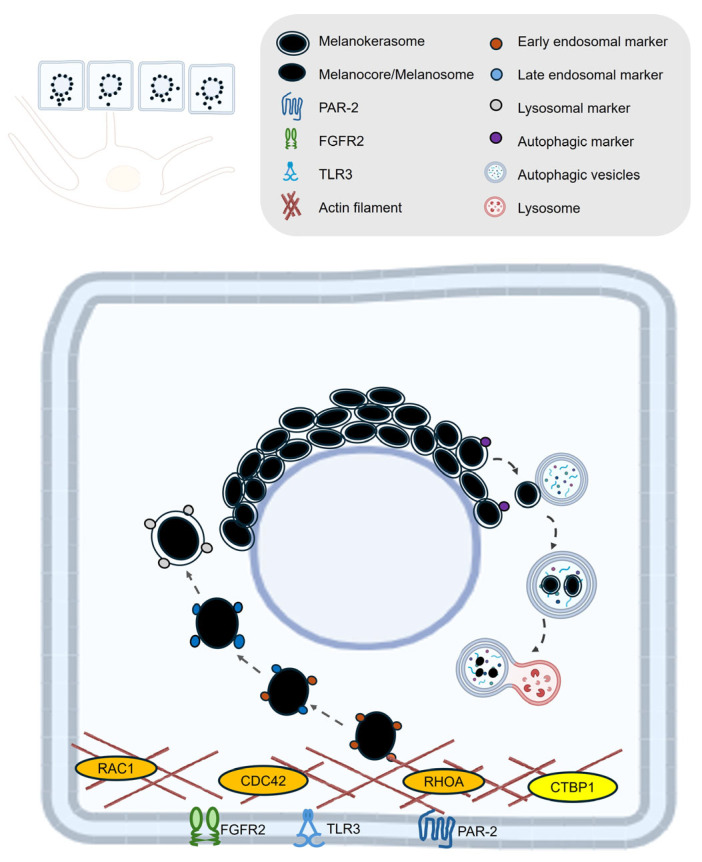
**Melanin Uptake, Retention and Degradation within Keratinocytes.** This schematic illustrates the process occurs within the keratinocyte region. The destiny of melanin uptake and processing in keratinocytes. It distinguishes between melanocore and melanosome internalization mechanisms, showing that: Melanocores are taken up primarily via phagocytosis, regulated by Rac1 and Cdc42, and mediated by PAR-2 activation. Melanosomes are internalized via macropinocytosis, involving CtBP1/BARS and RhoA, and are enhanced by KGF/FGF7-FGFR2b signaling and TLR3 activation under stress conditions. Once internalized, melanin-containing compartments are trafficked to the perinuclear region on the apical side of the cell, forming melanin caps that shield nuclear DNA. Vesicles mature through the endolysosomal pathway, transitioning from early endosomes (EEA-1, Rab5) to late endosomes/lysosomes (Rab7, CD63, LAMP1). Melanin is stored in melanokerasomes, which resist full degradation. Selective degradation occurs via autophagy, with p62, ATG7 and LC3 marking the pathway. Fusion with lysosomes, mediated by Rab7B/42, forms autolysosomes, where cathepsin V facilitates partial melanin breakdown. This process reflects a finely tuned balance between melanin retention for photoprotection and degradation for pigment homeostasis. The schemes were created in BioRender. Bolis, M. (2025) https://BioRender.com/h42n006.

**Table 1 ijms-26-08630-t001:** (**a**) Selected molecular biomarkers of skin pigmentation in melanocytes. (**b**) Selected molecular biomarkers of skin pigmentation in keratinocytes.

Process	Gene ID	Function
(**a**)		
**Melanogenesis**	*TYR*	Catalyzes the conversion of the amino acid tyrosine into melanin through a series of biochemical reactions [4].
	*TYRP1*	Catalyzes the oxidation of 5,6-dihydroxyindole-2-carboxylic acid (DHICA) to indole-5,6-quinone-2-carboxylic acid in the melanin biosynthesis pathway [4].
	*DCT*	Catalyzes the conversion of DHICA during melanin synthesis [4].
	*MITF*	A transcription factor that controls the expression of numerous genes involved in melanin synthesis and pigmentation [24].
	*MC1R*	A receptor activated by α-MSH that activates the cAMP signaling pathway, crucial for stimulating melanin production [23].
	*NRF3*	A transcription factor that regulates the uptake of melanin precursors, such as L-tyrosine and L-DOPA, through macropinocytosis and also controls the expression of autophagy-related genes involved in melanosome formation and degradation [31].
	*PMEL*	Initiates the formation of melanosome [32].
	*MART1*	Forms a complex with PMEL, thereby regulating PMEL’s expression, stability, trafficking, and proteolytic processing [34].
	*OA1*	Functions as a key regulator of melanosome maturation by controlling melanosome biogenesis and size at distinct stages [36,38].
	*OCA2*	Encodes a melanosomal membrane protein that contributes to a chloride ion current, which is essential for regulating melanosomal pH [43]
	*SLC45A2*	Encodes a melanosomal membrane transporter that functions at the late stages of melanosome maturation to maintain a neutral pH within mature melanosomes [46].
	*ATP7A*	A copper transporter that localizes to melanosomes in a BLOC-1–dependent manner, where it supplies copper directly to TYR [48].
**Transport and transfer**	*RAB1A*	A small GTPase that promotes melanosome microtubule anterograde transport [51].
	*SKIP (PLEKHM2)*	An adaptor protein that forms a transport complex with Rab1A and kinesin-1 to facilitate melanosome microtubule anterograde transport [51].
	*KIF5B*	The kinesin-1 heavy chain that regulates melanosome microtubule anterograde transport [51].
	*KCL2*	The kinesin-1 light chain that regulates melanosome microtubule anterograde transport [51].
	*MAP1LC3B*	Induces MITF expression, mediates melanosome-microtubule interactions to facilitate melanosome trafficking on microtubule and helps to translocate melanosome from microtubule to actin [14,58].
	*ATG4B*	Removes LC3B from microtubule and further mediates melanosome trafficking on actin [14,58].
	*MACF1*	Functions as a cytoskeletal crosslinker that coordinates the interaction between microtubules and actin filaments [60,61].
	*RAB27A*	A small GTPase that promotes melanosome actin transport [63,65].
	*Melanophilin*	An adaptor protein that bridges Rab27A/Myo5A and promotes melanosome actin transport [64,65].
	*MYO5A*	Functions as a processive actin-based motor protein that is essential for the short-range transport and peripheral capture of melanosomes in melanocytes [65,66].
	*RAB36*	Promotes melanosome microtubule retrograde transport [71].
	*RILP*	Interacts with Rab36 and promotes melanosome microtubule retrograde transport [71].
	*Melanoregulin*	Interacts with RILP and DCTN1 and mediates melanosome microtubule retrograde transport [68].
	*DYNLT3*	A regulatory subunit of the cytoplasmic dynein motor complex, specifically influencing melanosome retrograde transport in melanocytes [68,69].
	*RAB7A*	Promotes early-stage melanosome microtubule retrograde transport [72].
	*RAB44*	Promotes mature melanosome microtubule retrograde transport [70].
	*MYO10*	Is upregulated by ultraviolet radiation and Ca^2+^ stimulation and is important for filopodia formation and melanin transfer [81].
	*RAB17*	Is required for melanocyte filopodia formation and thereby facilitates pigment transfer [82].
	*RAB3A*	Regulates melanin exocytosis, particularly under stimulation by soluble factors from differentiated keratinocytes [86].
	*RAB11B*	Regulates keratinocytes induced melanin exocytosis and transfer [84,85].
	*EXOC7*	The subunits of the exocyst complex and is involved in melanin exocytosis and transfer [85].
	*EXOC4*	The subunits of the exocyst complex and is involved in melanin exocytosis and transfer [85].
	*CAV1*	Forms caveolae structures that facilitate melanocyte–keratinocyte interactions necessary for melanin transfer [80].
**Process**	**Gene ID**	**Function**
(**b**)		
**Uptake**	*PAR-2*	Activates phagocytic capacity of keratinocytes, receptor, promotes melanocore and melanosome uptake [10,87,88,93,94].
	*TLR3*	UV-responsive regulator of melanin internalization. Enhances melanosome and melanocore uptake in keratinocytes via actin-dependent endocytosis, primarily by activating RhoA and Cdc42 [98].
	*FGFR2*	Promotes melanosome uptake through phagocytosis and links this process to autophagy, controlling both the internalization and degradation of melanosomes in keratinocytes [95,97].
	*RAC1*	A Rho GTPase that mainly promotes melanocore uptake [87].
	*CDC42*	A Rho GTPase that mainly promotes melanocore uptake [87].
	*RHOA*	A Rho GTPase that mainly promotes melanosome uptake [87].
	*CTBP1*	Encodes a protein involved in membrane fission events necessary for endocytosis, particularly affecting melanosome uptake [87].
**Retention and degradation**	*LAMP1*	Regulates lysosomal exocytosis, a process critical for melanosome transport and integration into keratinocytes. Maintains lysosomal membrane integrity, protecting against enzymatic degradation and enabling melanin’s long-term photoprotective storage in keratinocytes [5,9,13,99].
	*EEA1*	Early endosomal marker that surrounds melanocores in keratinocytes [5,10].
	*RAB5*	Early endosomal marker that surrounds melanocores in keratinocytes [5,10].
	*P62*	Functions as an autophagy adaptor protein in keratinocytes, mediating the selective degradation of melanosomes by linking them to the autophagy machinery and facilitating their clearance through the autophagy–lysosome pathway [11,100].
	*ATG7*	Essential for autophagy-dependent melanosome degradation in keratinocytes by enabling the formation of autophagosomes that engulf and facilitate the lysosomal breakdown of melanin-containing compartments [11].
	*MAP1LC3B*	LC3 (specifically LC3-II, the lipidated form of MAP1LC3B) functions in melanosome degradation in keratinocytes by marking autophagosomes that engulf melanin-containing compartments, thereby facilitating their autophagic clearance through the lysosomal pathway [11].
	*RAB7B*	Facilitates lysosomal fusion and protein degradation on melanosomes [13].
	*CTSV*	Lysosomal protease plays a critical role in breaking down melanosome-associated proteins and melanosome integrity, indirectly influencing melanin persistence in keratinocytes [12].

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
