# Peer review of "Melanosome Transport and Processing in Skin Pigmentation: Mechanisms and Targets for Pigmentation Modulation"

_ijms, 2025, doi:10.3390/ijms26178630_

Round 1
Reviewer 1 Report
Comments and Suggestions for Authors
Although there are several recent reviews on the same subject, this one focuses on an interesting angle: the use of regulators of melanosome transport and melanin transfer/processing as targets to modulate skin pigmentation. It is generally well written and thorough, with useful schematics and tables, even though there are several aspects that can be improved:
1 – The title refers “emerging biomarkers”, which is odd and seems out of context with the content of the manuscript. Therefore, it should be replaced, for example, by “targets to modulate skin pigmentation” (or something similar).
2 – A general comment is that the review is acritical of the evidence found in the literature. This is more striking when the authors refer and explain the models of transfer proposed. Although there are 4 different ones, the amount and physiological relevance of the evidence for each one of them is very different. Therefore, authors should be (more) critical of the evidence included in the review.
3 – Even though the review is thorough, there are several regulators of melanogenesis (e.g., Rab4a, Rab6, Rab9, Rab22a, Rab32/38, AP-1/3, SLC24A5, among others), melanosome transport (Rab7a/dynein) and melanin secretion/transfer (Rab3a, Rab17, Myosin X, among others) that are not referred. Of course, it might not be possible to cover everything about all the processes, but the authors must state what were the criteria adopted to select the regulators they discuss.
Other (minor) comments:
Line 37 – “captured” might not be the best word and “tethering” is preferable.
Line 47 – “melanophagy” is used to refer to the autophagy of melanosomes inside melanocytes. Even though autophagy was shown to have a role in the degradation of melanokerasomes, the concepts should not be mixed or confused.
Line 108 – “series” should replace “serious”.
Line 148 – The title of Fig. 1 should be revised (suggestion: Melanin synthesis and melanogenesis steps)
Line 299 – The phagocytosis of melanocores is not dependent on RhoA.
Line 323 – References should be added to the sentence.
Line 324 – Sentence should be corrected to “...transported and positioned above...”.
Line 326 – The existence of tethers between the membrane of melanokerasomes and the nuclear envelope could be added.
Line 395 – The names of the membrane proteins should be added.
Finally, references should be corrected as there are details missing (see for example #2, 3, 5, etc.).
Reviewer 2 Report
Comments and Suggestions for Authors The article provides a relatively comprehensive review of the synthesis, transport, transfer, and metabolism processes of melanin. Particularly commendable is the summary of key biomarkers and their functions in these related processes, which offers important references for research on skin melanin. However, in my personal opinion, there are still some deficiencies or questions in the article, as follows: 1. The research results mentioned in the article regarding melanin transfer and transport show a high degree of consistency. Nevertheless, the Abstract of Reference 5 states, "Over the past few decades, distinct models have been proposed to explain how melanin transfer occurs at the cellular and molecular levels. However, this remains a debated topic, as up to four different models have been proposed, with evidence presented supporting each." Please confirm whether there are still hypotheses in different directions or studies with inconsistent conclusions up to now. If so, they should be mentioned in the article. 2. It is suggested that the classification of biomarkers in Table 1b should be consistent with that in Section 4 of the article. For example, "transfer and uptake" may be changed to "uptake", and "processing" may be changed to "retention and degradation".
Reviewer 3 Report
Comments and Suggestions for Authors
1. The figures in the manuscript are visually appealing; however, they could be enhanced by incorporating more detailed molecular interaction mechanisms, such as phosphorylation sites and conformational changes.
2. What are the main differences in the melanosome degradation pathway between humans and mice?
3. Should the terms "melanin nucleus" and "melanosome" be used more consistently throughout the article?
Round 2
Reviewer 1 Report
Comments and Suggestions for Authors
The authors reviewed the manuscript according to the comments and suggestions provided. This resulted in a more comprehensive and useful review for the readership. Final revisions of the added text should be made to improve English and flow (e.g., line 57 - "... processing processes." and the sentence in line 254, which should be rephrased). Moreover, the title of Fig. 1 is still not according to the other figure titles and the part "This schematic depicts the ..." should be deleted. Finally, just as an example, reference #5 is missing the second author (did not check further but all should be rechecked).
